# TGFα Promotes Chemoresistance of Malignant Pleural Mesothelioma

**DOI:** 10.3390/cancers12061484

**Published:** 2020-06-06

**Authors:** Bernard Staumont, Majeed Jamakhani, Chrisostome Costa, Fabian Vandermeers, Sathya Neelature Sriramareddy, Gaëlle Redouté, Céline Mascaux, Philippe Delvenne, Pascale Hubert, Roghaiyeh Safari, Luc Willems

**Affiliations:** 1Molecular and Cellular Epigenetics (GIGA) and Molecular Biology (GxABT), University of Liège, 4000 Liège, Belgium; b.staumont@uliege.be (B.S.); majeed.jamakhani@uliege.be (M.J.); costachrisostome@yahoo.fr (C.C.); fabian1508@hotmail.com (F.V.); sathy.ns@gmail.com (S.N.S.); gaelle.redoute@gmail.com (G.R.); roghaiyeh.safari@gmail.com (R.S.); 2Department of Pulmonology, Strasbourg University Hospital, 67091 Strasbourg, France; celine.mascaux@chru-strasbourg.fr; 3Interface de Recherche Fondamentale et Appliquée en Cancérologie, INSERM U1113, Université de Strasbourg, 67200 Strasbourg, France; 4Laboratory of Experimental Pathology, GIGA-Cancer, University of Liège, 4000 Liège, Belgium; p.delvenne@ulg.ac.be (P.D.); P.Hubert@uliege.be (P.H.)

**Keywords:** mesothelioma, TGFα, chemoresistance, combination therapy

## Abstract

*Background*: There is no standard chemotherapy for refractory or relapsing malignant pleural mesothelioma (MPM). Our previous reports nevertheless indicated that a combination of an anthracycline (doxorubicin) and a lysine deacetylase inhibitor (valproic acid, VPA) synergize to induce the apoptosis of MPM cells and reduce tumor growth in mouse models. A Phase I/II clinical trial indicated that this regimen is a promising therapeutic option for a proportion of MPM patients. *Methods*: The transcriptomes of mesothelioma cells were compared after Illumina HiSeq 4000 sequencing. The expression of differentially expressed genes was inhibited by RNA interference. Apoptosis was determined by cell cycle analysis and Annexin V/7-AAD labeling. Protein expression was assessed by immunoblotting. Preclinical efficacy was evaluated in BALB/c and NOD-SCID mice. *Results*: To understand the mechanisms involved in chemoresistance, the transcriptomes of two MPM cell lines displaying different responses to VPA-doxorubicin were compared. Among the differentially expressed genes, transforming growth factor alpha (TGFα) was associated with resistance to this regimen. The silencing of TGFα by RNA interference correlated with a significant increase in apoptosis, whereas the overexpression of TGFα desensitized MPM cells to the apoptosis induced by VPA and doxorubicin. The multi-targeted inhibition of histone deacetylase (HDAC), HER2 and TGFα receptor (epidermal growth factor receptor/EGFR) improved treatment efficacy in vitro and reduced tumor growth in two MPM mouse models. Finally, TGFα expression but not EGFR correlated with patient survival. *Conclusions*: Our data show that TGFα but not its receptor EGFR is a key factor in resistance to MPM chemotherapy. This observation may contribute to casting light on the promising but still controversial role of EGFR signaling in MPM therapy.

## 1. Introduction

Malignant mesothelioma is a highly aggressive neoplasm affecting the pleura, the peritoneum, the pericardium and the tunica vaginalis testis [1]. Malignant pleural mesothelioma (MPM) is the predominant type (75% of cases), while the peritoneal form represents about 20% of cases [2]. The causal link between exposure to asbestos and the development of MPM has been known for more than 50 years [3,4]. Because of the widespread use of asbestos-containing materials over the last decades, MPM presently has a worldwide distribution. The prognosis for patients with MPM remains poor, notably due to the advanced stage of the disease at presentation and because of resistance to available treatments. Therapeutic approaches have been disappointing, and the treatments used have not proven their ability in significantly prolonging survival in comparison to supportive care. The possibility of curative resection is extremely rare, and the impact of chemotherapy on the outcome of patients with mesothelioma is still limited. Two randomized studies have demonstrated an increase in the response rate and survival when comparing cisplatin and an antifolate (pemetrexed or raltitrexed) versus cisplatin alone [5,6]. The standard first-line chemotherapy therefore consists of a combination of cisplatin (or carboplatin) and pemetrexed (or raltitrexed). However, most patients become resistant to this treatment and relapse rapidly, with a median overall survival not exceeding 12.7 months [5,7]. For these refractory or relapsing subjects, there is, unfortunately, no standard second-line option. Among the available chemotherapeutic agents, doxorubicin is one of the most frequently used compounds [8,9,10,11]. Doxorubicin stabilizes the topoisomerase II complex after the dissociation of the DNA replication fork and thereby interferes with DNA synthesis [12]. However, most second-line monotherapies, including doxorubicin as a single agent, yield low response rates in MPM and do not significantly increase patient survival [13].

With the aim to improve the efficacy of second-line therapy, we hypothesized that MPM chemoresistance results from epigenetic errors associated with an inappropriate transcriptional profile. Among epigenetic modulators, we used a histone deacetylase (HDAC) inhibitor, valproate (VPA, sodium salt of 2-propylpentanoic acid), a short-chain fatty acid and a potent anti-convulsant widely used for decades to treat epilepsy and bipolar disorders [14]. The main advantages of VPA include the knowledge of its pharmacological properties and its good clinical tolerability at physiological concentrations [15,16]. VPA stimulates histone acetylation, modifies the gene expression profile and induces apoptosis in MPM cells in the presence of cisplatin and pemetrexed [17]. VPA also potentiates the antitumor effect of doxorubicin in cells and mouse models. Based on this preclinical evidence, a Phase II trial revealed a partial response rate of 16% and a disease control rate of 36%, indicating that the VPA-doxorubicin combination is a therapeutic option for second-line MPM patients [10]. Based on these promising observations, this study aims at further improving the efficacy of the second-line VPA-doxorubicin regimen.

## 2. Results

### 2.1. TGFα is Overexpressed in MPM Cells Resisting VPA-Doxorubicin Induced Apoptosis

In order to evaluate the mechanisms associated with the response of MPM to chemotherapy, the viability of different cell lines (M14K, M38K, ZL34 and H28) in response to the VPA HDAC inhibitor and/or the doxorubicin anthracyclin was compared. It appeared that the metabolic activity of all four cell types was moderately affected in the presence of VPA and/or doxorubicin (Appendix A). There was, nevertheless, a clear difference in the apoptotic rates as measured by phosphatidylserine exposure through Annexin V labeling. In contrast to M14K, M38K and ZL34, H28 cells were unresponsive to VPA and doxorubicin (Appendix A). We further analyzed DNA fragmentation in M14K and H28 characterized by a difference in sensitivity to doxorubicin-VPA treatment. Cell cycle analysis confirmed that a combination of doxorubicin (100 nM) and VPA (2 mM) was able to significantly induce apoptosis in M14K cells but not in H28 cells (Figure 1a). For simplification, these two cell lines will further be referred to as “sensitive” (M14K) and “resistant” (H28) to the chemotherapy.

To identify the mechanisms underlying this phenotype, the transcriptomes of H28 and M14K cells were analyzed by RNA sequencing (Illumina HiSeq 4000). The reads were mapped with the STAR software to the human genome (GRCh38, Ensembl). To identify the pathways involved, gene set enrichment analysis (GSEA) was performed with DESeq2 on the whole set of expressed genes ranked by their differential expression in the two cell lines (Appendix A). The results show that the gene set associated with the regulation of epidermal growth factor activated receptor (EGFR) activity was significantly enriched in H28 cells (*p* = 0.02; Figure 1b–c). The volcano plot based on log2 (fold change) and −log10 (false discovery rate) identified a list of genes that were differentially expressed in M14K and H28 cells (Figure 1d). With a log2 (fold change) greater than 6, TGFα was among the top overexpressed genes in H28 cells (blue arrow). Of note, the multidrug resistance pathway was enriched neither in M14K nor in H28 cells (Appendix A).

The difference in TGFα gene expression was validated by RT-qPCR (Figure 1e) and confirmed at the protein level by an ELISA performed on total cell lysates (Figure 1f). TGFα expression and chemosensitivity were further assessed in 10 mesothelioma cell lines. The Pearson correlation test was performed between the log10-transformed data of basal TGFα expression and chemosensitivity to doxorubicin-VPA. The analysis unveiled a negative correlation (Pearson correlation coefficient r = −0.6131) between the TGFα mRNA level and apoptosis induced by doxorubicin-VPA (Figure 1g). The values of apoptosis and expression levels for each cell line and their associated histologic type can be found in Appendix A.

### 2.2. Modulation of TGFα Expression Influences the Apoptotic Response Induced by Doxorubicin and VPA

We next evaluated the role of TGFα in resistance to the doxorubicin-VPA regimen. TGFα expression was negatively (Figure 2a) or positively (Figure 2b) modulated by RNA interference or gene transduction, respectively. TGFα transcription was quantified by RT-qPCR (left panels of Figure 2), while the apoptotic response to doxorubicin-VPA was evaluated by an Annexin V assay (right panels of Figure 2). The inhibition of TGFα expression sensitized H28 cells to doxorubicin-VPA (*p*-value < 0.01) (Figure 2a). Similarly, an increase in apoptosis was also observed in the chemosensitive M14K cells when TGFα expression was reduced (Figure 2a). Conclusions were validated independently by using another shRNA vector (Appendix A). Conversely, increasing the expression of TGFα led to a significant decrease in the apoptosis induced by doxorubicin-VPA in M14K cells (Figure 2b). Apoptotic levels remained low in H28 cells overexpressing TGFα. Similar results were obtained with the DNA fragmentation test (data not shown). Together, these results thus correlate resistance to doxorubicin-VPA chemotherapy with the levels of TGFα expression through a decrease in apoptosis.

### 2.3. EGFR Tyrosine Kinase Inhibitors Increase Apoptosis Induced by VPA and Doxorubicin

Since high TGFα expression correlates with resistance to doxorubicin-VPA and TGFα is a ligand of EGFR, we next inhibited EGFR signaling. For this perspective, two first-generation EGFR tyrosine kinase inhibitors (TKIs), gefitinib and erlotinib, were evaluated in combination with the doxorubicin-VPA regimen. The apoptotic response was measured in three mesothelioma cell lines (M14K, H28 and ZL34). As shown in Figure 3, both EGFR TKIs were unable to induce apoptosis when used as single agents. However, when combined with doxorubicin and VPA, gefitinib or erlotinib increased apoptosis in the three cell types. In particular, both EGFR TKIs could synergize with doxorubicin-VPA to significantly enhance apoptosis in chemoresistant H28 cells. Based on these observations, the therapeutic activity of this new regimen was evaluated in MPM mouse models. To this end, gefitinib or erlotinib, in combination with doxorubicin-VPA, were injected intraperitoneally (I.P.) to treat mice developing mesothelioma tumors. Unfortunately, the three-component regimen was not well tolerated in a series of mouse models due to excessive toxicity (data not shown).

### 2.4. An Inhibitor Targeting EGFR/HER2 and HDAC Synergizes with Doxorubicin to Induce Apoptosis in MPM Cells and Inhibit Tumor Growth in Two Mesothelioma Mouse Models

To overcome the problem of toxicity induced by a three-component treatment, we tested an inhibitor (CUDC-101) able to target class I–II HDACs, EGFR and HER2. The toxicity of CUDC-101 was evaluated by assessing the metabolic activity in H28, M14K and ZL34 cells (see dose response in the MTS assay in Appendix A). Based on this analysis and data from the literature [18,19], a subtoxic dose of 1 μM CUDC-101 was selected for further analysis. As expected, the phosphorylation of HER2 and EGFR, as well as the phosphorylation of EGFR downstream proteins (AKT and STAT1), were inhibited by CUDC-101 in H28 cells (Figure 4a). Furthermore, the acetylation of histone H3 was enhanced in the presence of CUDC-101. The intensity ratios as well as whole immunoblots can be found in Appendix A). As a single agent, CUDC-101 was unable to induce apoptosis but synergized with doxorubicin to significantly increase apoptosis in three MPM cell lines (H28, M14K and ZL34) (Figure 4b).

Based on the pro-apoptotic activity of CUDC-101 in cell lines, the therapeutic potential was evaluated in two mouse models. Murine AB12 and human M14K MPM cells were injected subcutaneously into BALB/c and NOD-SCID mice, respectively. Once the tumors had reached a mean size of 100 mm^3^, mice were treated with CUDC-101 (two times/week, I.P. at 100 mg/kg) or/and doxorubicin (one time/week, I.P. at 0.5 mg/kg). When used as single agents, neither CUDC-101 nor doxorubicin alone showed antitumor activity when compared to vehicle (Figure 4c). By contrast, the combination of CUDC-101 and doxorubicin was able to reduce tumor growth in both models. In the epithelial M14K subtype, tumor growth was almost completely inhibited. In the more aggressive AB12 biphasic model, the tumor size was significantly reduced.

### 2.5. Low TGFα Expression Predicts Patient’s Survival

To correlate TGFα gene expression with patient survival, datasets were downloaded from The Cancer Genome Atlas (TCGA) and used to generate Kaplan–Meier graphs (Figure 5). The optimal cutpoints between high and low expression levels of TGFα and EGFR were determined using maximally selected rank statistics (maxstat R package) in Figure 5a,b, respectively. MPM patients characterized by low TGFα expression had a better survival rate (*p* = 0.039, Figure 5c). Although not significant (*p* = 0.1), a similar trend was observed for EGFR expression (Figure 5d).

Survival times integrating TGFα gene expression (low or high) and therapeutic response were calculated from the TCGA dataset. As expected, the stratification of survival rates revealed that patients with partial/complete response with low TGFα expression survived for longer compared to those with stable/progressive diseases (Kaplan-Meier in Appendix A). Interestingly, there was no complete response in patients with high TGFα expression in the dataset. Compared to stable disease, a partial response was associated with longer survival in the high and low TGFα categories (Figure 5e). Importantly, low TGFα expression correlated with a better survival rate (Figure 5e).

## 3. Discussion

MPM is a very aggressive cancer of the pleura associated with poor prognosis. In first-line chemotherapy, the association of pemetrexed (or raltitrexed) with cisplatin shows the best response rate in first-line setting [5]. Nevertheless, MPM patients treated with this regimen relapse rapidly and most frequently become refractory to further therapeutic intervention. We previously proposed an approach based on the epigenetic modulation of gene expression combined with chemotherapy [17,20]. In particular, a clinical trial demonstrated that VPA-doxorubicin is a promising second-line therapy against MPM [10]. In this perspective, the present study aimed at further improving the clinical response to VPA-doxorubicin chemotherapy.

By comparing two MPM cell lines having different sensitivities towards VPA-doxorubicin, we identified TGFα as a key player in chemoresistance. TGFα is one of the seven human ligands that bind to the EGF receptor (EGFR or HER1). As a growth factor, TGFα is a signaling polypeptide involved in cell communication. Widely distributed in many tissues, TGFα plays an important role in cell homeostasis by stimulating survival, proliferation, tissue growth and the production of matrix components [21,22,23]. In the present study, we demonstrate that TGFα overexpression contributes to resistance to MPM chemotherapy. We show that TGFα expression negatively correlates with the apoptotic response to VPA-doxorubicin. Moreover, the inhibition of TGFα promotes apoptosis in poorly responsive H28 cells. Conversely, the overexpression of TGFα reduced the chemosensitivity of M14K cells. Collectively, these results support the major role played by TGFα in the resistance to VPA-doxorubicin therapy.

Our data show that EGFR inhibitors improve the therapeutic response to VPA and doxorubicin. In fact, EGFR is overexpressed in MPM as in other cancer types including breast cancer and non-small cell lung cancer [24,25]. EGFR can be targeted by monoclonal antibodies (cetuximab) and TKIs (gefinitib and erlotinib) [26]. Cetuximab induces potent antibody-dependent cellular cytotoxicity of MPM cells [27]. On the other hand, gefinitib (ZD1839) and erlotinib (OSI-774) were ineffective as single agents in Phase II clinical trials [28,29,30]. In combination, erlotinib potentiated the anti-cancer activity of arsenic [31]. Furthermore, the effect of MET or COX-2 inhibitors was improved by anti-EGFR TKIs in vitro [32,33]. In this report, we have shown that both gefitinib and erlotinib TKIs can act in synergy with the combination of VPA-doxorubicin to induce apoptosis in MPM cell lines. In particular, the new regimen sensitizes chemoresistant H28 cells to apoptosis, emphasizing the anti-cancer potential of EGFR inhibitors when used in combination therapies.

Unfortunately, the three-component treatment (VPA-doxorubicin-erlotinib or VPA-doxorubicin-gefitinib) induced excess toxicity in mouse models. We hypothesized that the intrinsic toxicity of VPA-doxorubicin-TKI in mice could be reduced by using a multi-targeted inhibitor, such as CUDC-101. This compound combines a hydroxamic acid structure found in HDAC inhibitors such as vorinostat and the quinazoline moiety of EGFR TKIs [34]. The two-component treatment doxorubicin-CUDC-101 was well tolerated and inhibited tumor growth in two mouse models of MPM. While combinations of EGFR and HDAC inhibitors have been reported, CUDC-101 is efficient as a single agent in several preclinical cancer models including anaplastic thyroid cancer, erlotinib-resistant glioblastoma, and lung, colon and breast cancers [35]. Clinical trials (NCT01702285, NCT00728793, NCT01171924 and NCT01384799 at ClinicalTrials.gov, https://clinicaltrials.gov/) revealed that CUDC-101 was well tolerated by patients with advanced solid tumors [36]. In our preclinical study, we show that CUDC-101 has anticancer potential when combined with a DNA-damaging agent such as doxorubicin. It is noteworthy that the effective dose of doxorubicin was particularly low compared to that reported in other studies, indicating a synergism with CUDC-101 [37,38]. The toxicity and efficacy of the CUDC-101 + doxorubicin regimen will have to be evaluated in further clinical trials with mesothelioma patients.

Besides a potential therapeutic outcome, this report also highlights the clinical relevance of TGFα based on a TCGA dataset of 87 mesothelioma patients. Indeed, low TGFα expression correlated with a better survival rate independently of the type of treatment (Figure 5). By contrast, the correlation with EGFR was not significant. It is noteworthy that seven ligands bind to and activate EGFR: TGFα, EGF, heparin-binding EGF-like growth factor, betacellulin, amphiregulin, epiregulin and epigen [39]. This multiplicity of ligands may explain why TGFα but not EGFR has potential prognostic value. In particular, EGF and TGFα have a distinct preference to produce EGFR/ERBB2 heterodimers compared with EGFR/EGFR homodimers. Differences in receptor signaling provoked by the two ligands are a potential basis for the heterogeneity in biological outcomes. The complexity of the mechanisms involved is further amplified by the ability of TGFα to act in a paracrine manner to modulate the tumor microenvironment [40].

The present study was initiated from a Phase 2 trial evaluating a second-line regimen combining VPA and doxorubicin [10]. This treatment was associated with toxicity (mainly leukopenia and neutropenia) and moderate activity (a partial response rate of 16% and a disease control rate of 36%). As observed in the mouse model, it is predicted that a combination of doxorubicin, VPA and EGFR/HER2 inhibitors may also benefit relapsing or unresponsive mesothelioma patients, providing that toxicity issues are controlled. This multidrug chemotherapy may indeed have immunosuppressive effects by affecting essential components of host immunity. On the contrary, the new regimen may induce immunogenic cell death and improve immunotherapy based on immune checkpoint inhibitors [41,42]. The rationale of this combinatorial approach is supported by accumulating evidence indicating that chemotherapy regulates the composition and function of tumor infiltrating lymphoid and myeloid cells [43,44]. The molecular mechanisms include the release of damage-associated molecular patterns (DAMPs), the activation of NFkB signaling, a higher PD-L1 expression on tumor cells, an increase in CD8+ T-cells, the maturation of antigen-presenting cells (APC), augmented antigen presentation through MHC-I and the downregulation of immunosuppressive cells at the tumor site (Treg and myeloid-derived suppressor cells). In this context, it is interesting that a decreased expression of TGFα correlates with the survival of patients undergoing different types of therapy (Figure 5c,e). Nevertheless, the potential value of any therapeutic combination should be clearly demonstrated, notably by better defining the patients who will benefit the most from these treatments.

## 4. Materials and Methods

### 4.1. Cell Culture and Chemicals

Human (NCI-H2452, H28, M14K, M38K, MSTO-211H, SPC111, SPC212, ZL5, ZL34 and ZL55) and murine mesothelioma cells (AB12, see Vandermeers et al. [17]) were cultured in RPMI 1640 (or in DMEM for M14K and M38K), supplemented with 10% fetal calf serum (FCS), 2 mM L-glutamine and antibiotics (penicillin-streptomycin). All the cell lines were maintained at 37 °C in a humidified atmosphere containing 5% CO_2_. The references for all human MPM cell lines can be found in Appendix A.

VPA (sodium salt of 2-propylpentanoic acid from Sigma-Aldrich, St. Louis, MO, USA) was dissolved in distilled water. Doxorubicin, EGFR tyrosine kinase inhibitors (gefitinib and erlotinib, Sigma-Aldrich, St. Louis, MO, USA) and the multi-targeted (HDAC, EGFR and HER2) inhibitor CUDC-101 (MedchemExpress [18]) were solubilized in DMSO. The final concentration of DMSO in the culture medium did not exceed 0.1%.

### 4.2. RNA Sequencing and Bioinformatics Analyses

RNA was isolated from M14K and H28 cells using the RNeasy Mini Kit (QIAGEN, Hilden, Germany) and then treated with DNase with a DNA-free DNA Removal Kit (Thermo Fisher Scientific, Waltham, MA, USA). RNA integrity and quantity were evaluated with a bioanalyzer (Agilent Technologies, Santa Clara, CA, USA). When the RNA Integrity Number (RIN) was > 8, libraries were generated using the CATS mRNA-seq kit v2 with poly(A) selection (Diagenode, Liège, Belgium) following the manufacturer’s instructions. The size distribution of the libraries was monitored with a bioanalyzer (Agilent Technologies, Santa Clara, CA, USA). Sequencing of the libraries (1 × 50 bp) was performed on an Illumina HiSeq 4000 sequencing device (Illumina, San Diego, CA, USA). After quality control of the FASTQ data (FastQC, version 0.11.9, open source, available at https://www.bioinformatics.babraham.ac.uk/projects/fastqc/), adapter reads, reads shorter than 18 base pairs and low-quality reads were trimmed. Sequence reads were then mapped against the human reference genome, GRCh38 Ensembl, using STAR (free open source software, version 2.7) [45]. RNA-seq reads were counted and analyzed using the DESeq2 version 1.20.0 R package (open source) [46]. Gene set enrichment analysis (GSEA, free open source software package) was performed on the differential expression data using GSEA version 2.2.4 [47].

### 4.3. Lentiviral Production and Generation of Stable Cell Lines

The pCSEF-IB-TGFα expression plasmid was obtained by recombination (LR Clonase; Invitrogen) using pDONR223-TGFα and the pCSEF-IRES-bsd lentiviral vector. TGFα shRNAs cloned into the pLKO.1-puro lentiviral vector were provided by Sigma-Aldrich (Mission shRNA). Controls were pCSEF-IB-Ctrl (empty vector) and pLKO.1-shRNA-Ctrl (pLKO.1-puro Non-Target shRNA Control Plasmid DNA, Sigma-Aldrich, St. Louis, MO, USA).

Packaging psPAX2 (5 µg), pVSV-G (5 µg) and pLKO.1-shRNA or pCSEF-IB vectors (10 µg) were transfected into HEK 293T cells by calcium phosphate precipitation. After 72 h, lentivirus-containing media were recovered and used to transduce H28 and M14K MPM cell lines. Infected cells were selected over 2 weeks in the presence of puromycin (1 µg/mL) or blasticidin (5 µg/mL) and frozen in FCS-DMSO (90–10%).

### 4.4. Quantification of TGFα Expression by RT-qPCR and ELISA

Total RNA was extracted from cultured cells using the NucleoSpin RNA kit (Macherey-Nagel, Düren, Germany). Complementary cDNAs were synthesized with a random primer mix using the ProtoScript II First Strand cDNA Synthesis Kit (New England Biolabs, Ipswich, MA, USA). Then, the cDNAs were PCR-amplified with primers specific for HPRT (5^′^-GGTCAAGAAGCATAAACCAAAG-3^′^ and 5^′^-AAGGGCATATCCCACAACAAAC-3^′^) or TGFα (5^’^-GCCCGTAAAATGGTCCCCTC-3^′^ and 5^′^-GACGTGCTGTTCTCCAAGGC-3^′^). Amplification was performed in a Roche Light Cycler using the Takyon SYBR master mix dTTP blue (Eurogentec, Liège, Belgium) according to the following protocol: 5 min denaturation at 95 °C followed by 45 cycles (15 s 95 °C, 20 s 60 °C, 40 s 72 °C) and termination, with a melting curve. Relative TGFα/HPRT expression was calculated using the Delta-Delta Ct method.

Human TGFα protein was quantified with the Quantikine Immunoassay according to the manufacturer’s protocol (R&D Systems, Minneapolis, MN, USA).

### 4.5. Quantification of Apoptosis

Cells were treated with 2 mM VPA, 100 nM doxorubicin, 3 µM gefitinib, 3 µM erlotinib and/or 1 µM CUDC-101 for 48 h. Apoptosis was determined by quantifying DNA fragmentation by flow cytometry (i.e., sub-G1 peak). Cells were fixed with 70% chilled ethanol and labeled with propidium iodide (20 µg/mL) after an RNAse (50 µg/mL) treatment. Following flow cytometry and cell distribution analysis, apoptosis was measured based on the sub-G1 cell population. The second method used to quantify apoptosis was based on the detection of phosphatidylserine exposure with the PE Annexin V Apoptosis Detection Kit (BD Biosciences, Franklin Lakes, NJ, USA). Twelve thousand events were selected by flow cytometry (FACS Canto II, BD Biosciences) and analyzed with the FACS Diva Software (version 6.1.2, BD Biosciences, Franklin Lakes, NJ, USA). Annexin V+/7-AAD- and Annexin V+/7-AAD+ cells were considered as early and late apoptotic cells respectively.

### 4.6. Cell Lysate Preparation and Immunoblotting

MPM cells were cultured for 4 h with CUDC-101 (1 µM) and/or doxorubicin (100 nM). After stimulation with human Epithelial Growth Factor EGF (10 ng/mL) for 10 min, cells were washed with cold PBS and lysed on ice for 10 min in buffer LB-150 (50 mM Tris-Cl pH 7.4, 150 mM NaCl, 0.5% NP-40, 0.5 mM EDTA) supplemented with a protease and phosphatase inhibitor cocktail (Halt 100x, Thermo Fisher Scientific). Cells were scraped from the plates and further incubated for 20 min on ice. After centrifugation, cell lysate supernatants were recovered and stored at −80 °C.

Protein concentration was measured by a bicinchoninic acid (BCA) assay (Pierce BCA Protein Assay Kit, Thermo Fisher Scientific, Waltham, MA, USA). Bovine serum albumin (BSA) was used as calibration standard. Proteins (40 µg) were resolved by SDS-polyacrylamide gel electrophoresis (SDS-PAGE) and transferred onto a nitrocellulose membrane (Whatman Protran BA85, GE Healthcare Life Sciences, newly Cytiva, Marlborough, MA, USA). The membranes were incubated for 1 h with a blocking buffer 5% (w/v), BSA or nonfat dry milk in TBS (Tris-buffered saline) supplemented with 0.1% Tween20. Blotted proteins were labeled overnight at 4 °C with primary antibodies: phospho-EGF Receptor (Tyr1068 D7A5), phospho-HER2/ERBB2 (Tyr1248), HER2/ERBB2 (D8F12), acetyl-Histone H3 (Lys9/Lys14) and heat-shock protein 90 (from Cell Signaling Technology, Danvers, MA, USA) and phospho-STAT1 (Tyr701) and phospho-AKT (Thr308) (from Abcam, Cambridge, UK). The membranes were then washed with TBS-Tween20 and incubated with appropriate anti-immunoglobulin horseradish peroxidase conjugates (HRP, Dako, newly Agilent Technologies, Santa Clara, CA, USA) for 1 h at room temperature. After washing with TBS-Tween20, the membranes were developed with a chemiluminescent HRP substrate (Pierce ECL Western Blotting Substrate, Thermo Fisher Scientific, Waltham, MA, USA) using a cooled CCD camera (ImageQuant LAS4000 mini, GE Healthcare Life Sciences, newly Cytiva, Marlborough, MA, USA).

### 4.7. Mouse Models

Studies were performed in an accredited laboratory of the University of Liège (LA1610002) in accordance with European Union standards. Animal experimentation was approved by the Ethics Committee for the use of laboratory animals at the University of Liège (case number 13-1564). BALB/c and NOD-SCID mice (provided by animal facility LA2610359) were inoculated subcutaneously into both the right and left flanks with 1 million to 4 million cells (sarcomatoid AB1 in BALB/c mice and epithelial M14K in NOD-SCID mice). A total of 200 µL of medium suspension containing 50% (*v*/*v*) Matrigel (Basement Membrane Matrix, Corning, Corning, NY, USA) was injected in each flank through a 27G needle. Once the average tumor volume reached 100 mm^3^, the mice were randomized into four groups (*n* = 6) in order to minimize weight and tumor size differences. The mice were mock-treated (with 30% Captisol solution, Ligand, San Diego, CA, USA) or injected intraperitoneally with CUDC-101 dissolved in 30% Captisol (twice per week at 100 mg/kg) and/or doxorubicin in water (once per week at 0.5 mg/kg). Tumors were measured bi-weekly with a digital caliper, and tumor volume was estimated by using the formula (π × length × width^2^)/6.

### 4.8. Analysis of the Cancer Genome Atlas Database

Gene expression data and clinical information were downloaded from the TCGA-MESO dataset (https://www.cancer.gov/tcga) through the Genomic Data Commons (GDC) by using TCGAbiolinks of the R package [48]. The gene expression data and clinical information were merged according to the patient’s barcode and combined in a matrix for the analysis of survival times by using open source R survMiner (version 0.4.0). Patients were classified into high and low expression groups using the R maxstat (open source R package, version 0.7-25), which predicts the optimal cutpoint for TGFα and EGFR expression. High and low expression groups were combined with survival rates to generate a Kaplan-Meier graph. The log rank *p*-value with a minimum threshold of 0.05 was calculated by using the R survMiner package. The response to treatment was included into a matrix created according to patient’s barcode. Four categories, i.e., stable or progressive disease, and partial and complete responses, were combined with gene expression data for survival analysis using R survMiner.

### 4.9. Statistical Analysis

The data collected were analyzed using the R software (version 3.6.1). Statistical significance was calculated using the Mann-Whitney U test (Wilcoxon rank-sum test), independent Student’s t-test or one-way analysis of variance (ANOVA), as indicated. The Pearson correlation test was used to evaluate the relationship between the TGFα expression level and apoptosis induced by VPA–doxorubicin. Data were considered significantly different when *p*-values < 0.05 (*), < 0.01 (**) and < 0.001 (***).

## 5. Conclusions

Our report indicates that TGFα is a key factor in the resistance of MPM to doxorubicin, a drug that can be used in second-line therapy for this disease. This observation may contribute to casting light on the promising but still controversial role of EGFR signaling in MPM therapy. Whether TGFα expression can be used as a prognostic factor or a therapeutic target needs to be further validated in prospective trials.

## Figures and Tables

**Figure 1 cancers-12-01484-f001:**
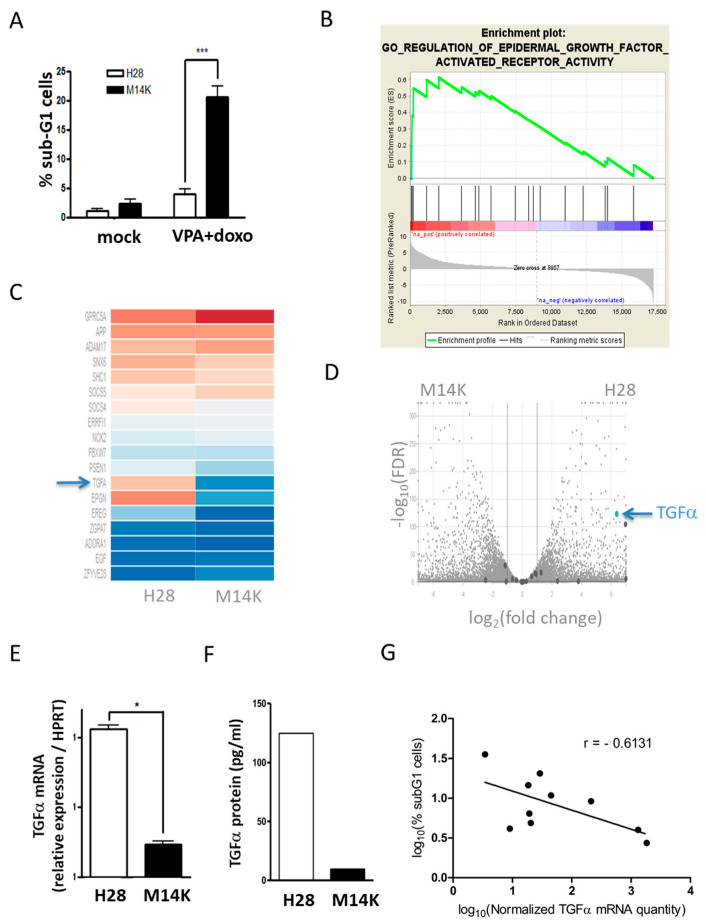
The sensitivity of mesothelioma (MPM) cells to valproic acid (VPA)-doxorubicin chemotherapy negatively correlates with expression of TGFα. (**A**) H28 and M14K mesothelioma cells were treated for 24 h with VPA (2 mM) and doxorubicin (100 nM). Apoptotic DNA fragmentation was evaluated by flow cytometry after ethanol fixation and propidium iodide labeling. Data (% of sub-G1 cells) are means ± standard deviations (SD) of three independent experiments. *p*-value < 0.001 (***) was obtained by the independent Student’s t test. (**B**) RNA sequencing of H28 and M14K cells. Enrichment plot of epidermal growth factor activated receptor (EGFR) activity gene set determined by gene set enrichment analysis (GSEA). (**C**) Hierarchical heat map of genes belonging to GO regulation of EGFR activity gene set. (**D**) Volcano plot based on log2 (fold change) and −log10 (false discovery rate, FDR). (**E**) The difference in TGFα transcription was confirmed by RT-qPCR in H28 and M14K cells. The HPRT housekeeping gene was used as an endogenous control in order to normalize TGFα expression. RT-qPCR data are represented as means ± SD, and statistical significance was calculated using the Mann-Whitney U test (* *p*-value < 0.05). (**F**) Quantification of TGFα protein by ELISA. (**G**) A correlation test (Pearson) was performed between the log10-transformed data of normalized TGFα expression and the levels of apoptosis induced by VPA-doxorubicin in 10 mesothelioma cell lines (M14K, M38K, MSTO-211H, NCI-H2452, H28, SPC111, SPC212, ZL34, ZL5 and ZL55). Apoptosis was measured by quantifying the proportion of sub-G1 cells by flow cytometry. The results are means of three independent experiments, and the Pearson correlation coefficient (r) is represented on the graph (*p*-value = 0.0594).

**Figure 2 cancers-12-01484-f002:**
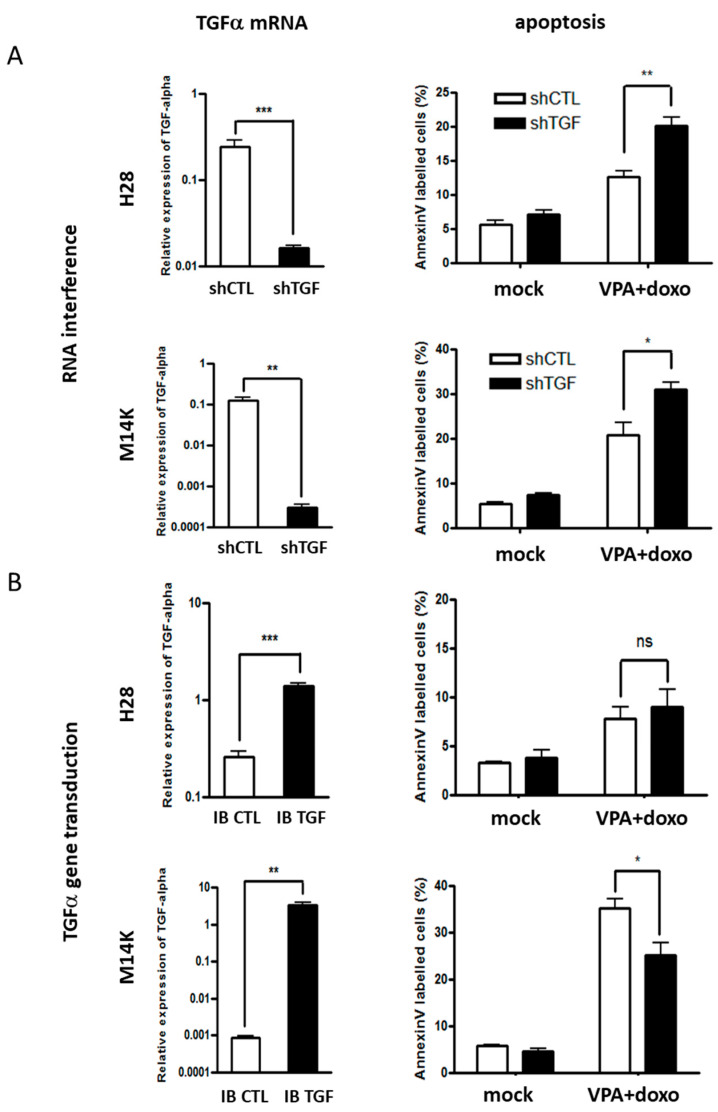
RNA interference and overexpression of TGFα increases and reduces the apoptosis induced by VPA and doxorubicin, respectively. (**A**) TGFα expression was inhibited in H28 and M14K cells by RNA interference using shRNA cloned into pLKO.1 (shTGF). A scrambled shRNA was used as control (shCTL). Downregulation or increased expression of TGFα was confirmed by RT-qPCR (left panels). The HPRT gene transcript was used as an endogenous control to calculate the relative expression of TGFα. Expression levels are represented as means ± SD of at least three independent experiments. Statistical significance was evaluated with the Mann–Whitney U test. The apoptosis induced by VPA-doxorubicin (VPA + doxo) was measured by an Annexin V assay (right panels). Apoptosis rates were calculated from three independent experiments and are represented as means ± SD. Means were compared using the independent Student’s t-test. * *p*-value < 0.05; ** *p*-value < 0.01; *** *p*-value < 0.001; ns: not significant. (**B**) TGFα was overexpressed in H28 and M14K cells using the plasmid pCSEF-IB-TGFα (IB TGF), while the empty vector pCSEF-IB served as control (IB CTL). TGFα expression and apoptotic rates were determined as described in panel **A**.

**Figure 3 cancers-12-01484-f003:**
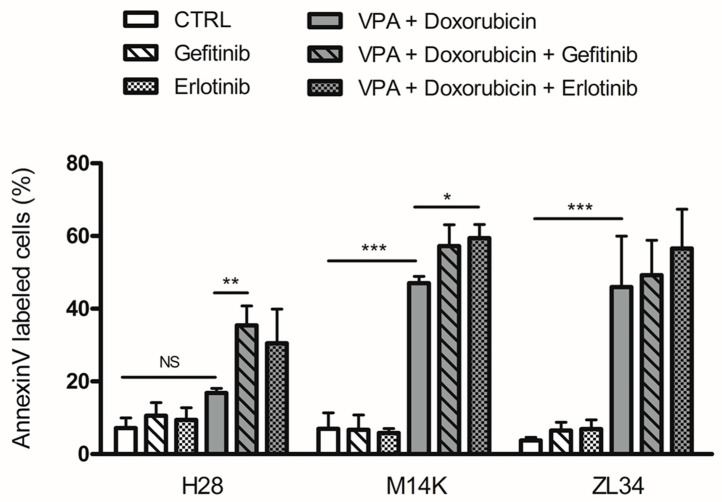
EGFR tyrosine kinase inhibitors increase the apoptosis induced by VPA and doxorubicin. H28, M14K and ZL34 cells were treated with VPA, doxorubicin and/or an EGFR tyrosine kinase inhibitor (gefitinib or erlotinib) and were evaluated for their apoptotic response by an Annexin V assay. Data are expressed as the means ± SD. Comparisons of multiple groups were performed using one-way analysis of variance (ANOVA) followed by Tukey’s multiple comparison procedure. Data are from three independent experiments. * *p*-value < 0.05; ** *p*-value < 0.01; *** *p*-value < 0.001; NS: not significant.

**Figure 4 cancers-12-01484-f004:**
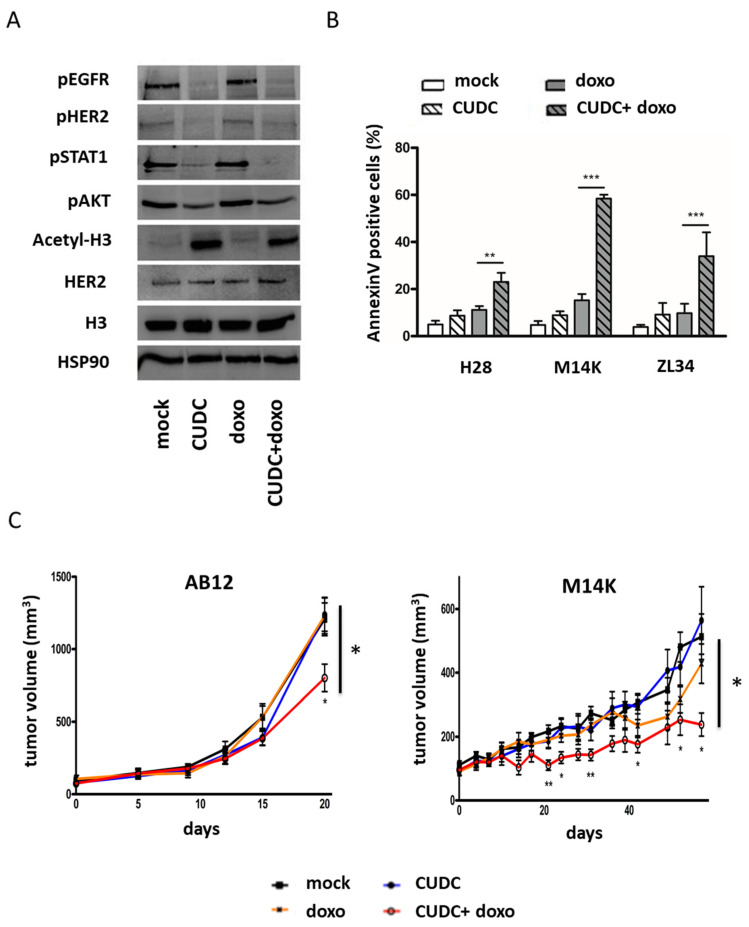
An inhibitor targeting EGFR/HER2 and HDAC synergizes with doxorubicin to induce apoptosis and inhibit tumor growth. (**A**) Representative immunoblotting of H28 cell lysates evaluating the phosphorylation of EGFR, STAT1, AKT and HER2 as well as the acetylation of histone H3 in response to doxorubicin and/or the EGFR/HER2/HDAC inhibitor (CUDC-101 at 1 µM). Heat-shock protein 90 (HSP90) was used as a loading control. (**B**) The apoptosis induced by CUDC-101 (1 µM) and/or doxorubicin (100 nM) was quantified by an Annexin V assay in three mesothelioma cell lines (H28, M14K and ZL34). Data represent the means ± SD of three independent experiments. Statistical significance was determined with one-way analysis of variance (ANOVA) followed by Tukey’s multiple comparisons test. ** *p*-value < 0.01; *** *p*-value < 0.001. (**C**) Tumor growth of AB12 (murine) and M14K (human) mesothelioma cells was evaluated after subcutaneous injection in BALB/c and NOD-SCID mice, respectively. Mice were treated intraperitoneally with mock (30% Captisol), doxorubicin (0.5 mg/kg once a week) or/and CUDC-101 (100 mg/kg twice a week). The tumor volume was measured at regular intervals. One-way analysis of variance (ANOVA) was used at the different time points to evaluate the statistical significance (* *p*-value < 0.05; ** *p*-value < 0.01) between the different treatment conditions (six mice per group).

**Figure 5 cancers-12-01484-f005:**
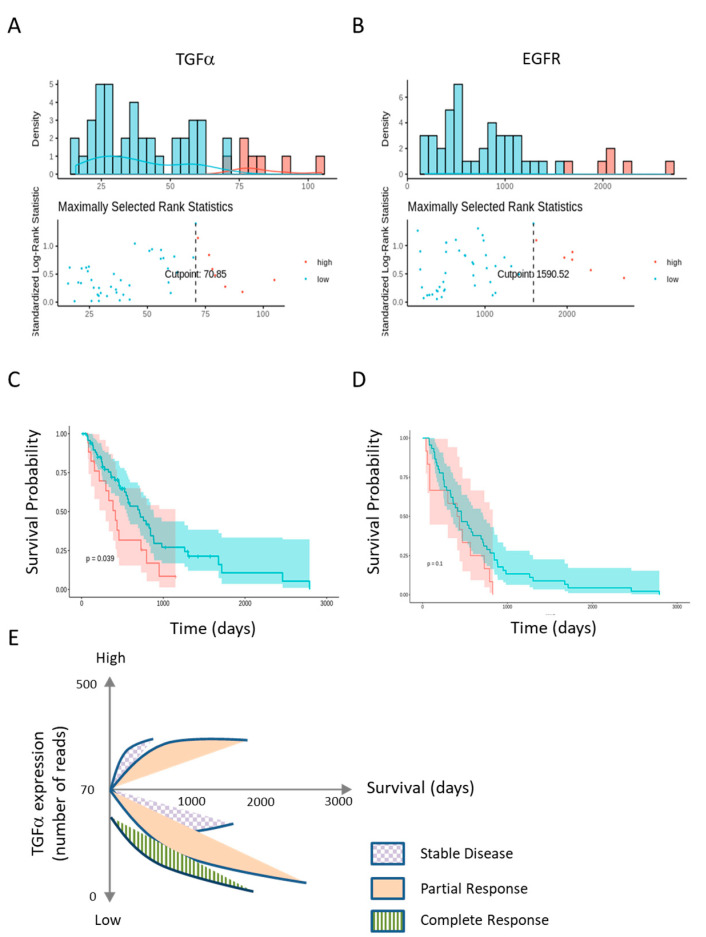
Kaplan-Meier survival curves of patients classified according to TGFα and EGFR expression. TGFα and EGFR expression datasets were downloaded from The Cancer Genome Atlas (TCGA). (**A**) and (**B**) Optimal cutpoints between high and low expression levels were calculated by maxstat for TGFα and EGFR, respectively; (**C**) and (**D**) Kaplan-Meier survival graphs were generated for patients categorized according to TGFα and EGFR expression; (**E**) Correlation of therapeutic response with survival of patients characterized by low and high TGFα expression.

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
