# Peer review of "TGFα Promotes Chemoresistance of Malignant Pleural Mesothelioma"

_cancers, 2020, doi:10.3390/cancers12061484_

Round 1

Reviewer 1 Report

The authors report on the results of a translational and in vitro research on the interference between the EGFR and a second line treatment with valproate and doxorubicin, an uncommonly used second line regimen with moderate activity and high toxicity, tested in a phase 2 trial . The research is nicely conducted and described and its results are interesting for the interaction at the EGFR level with TKI's but as mentioned by the authors, a combination of 3 drugs will be precluded by excessive toxicity.

These and other considerations should be reflected in the discussion, which should focus on future developments, e.g. the expected introduction of first line immunotherapy.

Author Response

   As requested, we added a paragraph in the discussion to emphasize the toxicity and efficacy of the VPA+doxorubicin treatment. We also extended the discussion on future immunotherapy developments.

   We added on line 301: …"The present study initiated from a phase 2 trial evaluating a second line regimen combining VPA and doxorubicin (10). This treatment was associated with toxicity (mainly leukopenia and neutropenia) and moderate activity (partial response rate of 16% and a disease control rate of 36%). As observed in the mouse model, it is predicted that a combination of doxorubicin, VPA and EGFR/HER2 inhibitors may also benefit relapsing or unresponsive mesothelioma patients providing that toxicity issues are controlled. This multidrug chemotherapy may indeed have immunosuppressive effects by affecting essential components of host immunity. On the contrary, the new regimen may induce immunogenic cell death and improve immunotherapy based on immune checkpoint inhibitors (41,42). The rationale of this combinatorial approach is supported by accumulating evidence indicating that chemotherapy regulates the composition and function of tumor infiltrating lymphoid and myeloid cells (43,44). The molecular mechanisms include the release of damage-associated molecular patterns (DAMPs), activation of NFkB signaling, a higher PD-L1 expression on tumor cells, the increase of CD8+ T-cells, the maturation of antigen-presenting cells (APC), augmented antigen presentation through MHC-I and downregulation of immunosuppressive cells at the tumor site (Treg and myeloid-derived suppressor cells). In this context, it is interesting that decreased expression of TGFα correlates with survival of patients undergoing different types of therapy (Figure 5C and E). Nevertheless, the potential value of any therapeutic combination should be clearly demonstrated, notably by better defining the patients who will benefit the most from these treatments …"

Reviewer 2 Report

Staumont et al. report that TGFα could be considered a key factor in mediating chemoresistance in Malignant Pleural Mesothelioma (MPM), and its overexpression seems to be correlated to a lesser sensitivity to doxorubicin/valproic acid (VPA) combined therapy, and not associated to EGFR mechanism of action.

This is an interesting paper which introduces a potential basis in better understanding the mechanism by which a so strong cancer, as MPM, is be able to resist to chemotherapy, currently the only possible therapeutic approach.

However, although it is a well conducted study, there are some problems as stated below and additional experiments are also needed to resolve these issues.

  1. Experiments shown by authors have been performed particularly in two MPM cell lines: M14K, used as cells “sensitive” to chemotherapy, and H28, as the “resistant” ones. This is in accordance to the histotype of these cell lines: M14K are epitheliod MPM cells, less aggressive tumoral cells and more responsive to drugs (as demonstrated widely in literature), while H28 are sarcomatoid MPM cells, much more refractory to chemotherapeutics. So, have authors evaluated and confirmed these differences also in the other MPM cell lines indicated in the paper? What are data concerning biphasic MPM cells (M38K, MSTO-211H, ZL5)?
  2. Moreover, have authors checked some markers of chemoresistance (i.e. MRP1) to confirm the distinction between “sensitive” and “resistant” cells? Authors should perform some experiments to evaluate at least one MDR marker, thus addressing this point.
  3. All the work was done in MPM cell lines: in my opinion, authors should evaluate the basal expression of TGFα also in not transformed mesothelial cells, to validate the hypothesis proposed. Moreover, it could be interesting to perform some experiments to clarify if an induced TGFα overexpression should be likely considered crucial, in these conditions, in promoting chemoresistance.

Author Response

Comment 1: "... have authors evaluated and confirmed these differences also in the other MPM cell lines indicated in the paper? What are data concerning biphasic MPM cells (M38K, MSTO-211H, ZL5)?..."

   As requested, we have evaluated the expression of TGFa in a series of cell lines and tested their response to VPA+doxorubicin treatment (Supplementary Table S2).

   The analysis reveals a negative correlation between log10-transformed data of basal TGFα expression and chemosensitivity to doxorubicin+VPA. (Pearson correlation coefficient r = - 0.6131). We added on line 106: “… Values of apoptosis and expression levels for each cell line and their associated histologic type can be found in Supplementary Materials (Additional file: Table S2) …”

Comment 2: "...have authors checked some markers of chemoresistance (i.e. MRP1) to confirm the distinction between “sensitive” and “resistant” cells? Authors should perform some experiments to evaluate at least one MDR marker..."

   To address this question, we analyzed the enrichment of the multidrug resistance pathway (BIOCARTA_MRP_PATHWAY) in transcriptomic data of M14K and H28 cells. This analysis shows that multidrug resistance pathway is enriched neither in M14K nor in H28 cells (p = 0.55). The distinction between “sensitive” and “resistant” cells thus appears not to be correlated with the multidrug resistance pathway.

   This is now indicated in supplementary Figure S3 and described on line 97: “… Of note, the multidrug resistance pathway was enriched neither in M14K nor in H28 cells (Additional file: Figure S3) …”

Comment 3: "... authors should evaluate the basal expression of TGFα also in not transformed mesothelial cells..."

   As requested, we analyzed expression of TGFα in normal mesothelial cells (Met5A) using Affymetrix Human U133 Plus 2.0 array data from GSE29370 data set available at GEO NCBI (Yoshikawa et al., 2011). Expression of TGFα was reduced in Met5A compared to H28.

See Figure A in attached pdf

Log2-transformed data of TGFα expression levels in normal mesothelial cells (Met5A) H28

Reference :
Yoshikawa Y, Sato A, Tsujimura T, Morinaga T et al. Frequent deletion of 3p21.1 region carrying semaphorin 3G and aberrant expression of the genes participating in semaphorin signaling in the epithelioid type of malignant mesothelioma cells. Int J Oncol 2011 Dec;39(6):1365-74.

Reviewer 3 Report

In the present manuscript entitled "TGFα promotes chemoresistance of malignant pleural mesothelioma” by Beranerd Staunmont et. al., authors investigated the efficacy of a combination regimen of doxorubicin plus valproic acid (VPA) in human Malignant pleural mesothelioma (MPM) cell lines. Notably, they found that H28 cell line was more resistant to this therapy compared to the other cell lines, at least in part due to high expressing level of TGFα. Ablation of the latter has sensitized H28 cells to the combined regimen of doxo+VPA. About that, It could be helpful, for better understand the role of TGFα in the mesothelioma acquired resistance, to generate a doxo+VPA-resistante cell line. After that, authors should tested in this experimental condition their hypothesis, as well as checked whether a tgf ablation in this resistante cellular model could revert the acquired resistance.

Basing on these data, authors suggested that TGFα is a critical protein for mesothelioma chemo resistance acquisition and proposed to overcome its pro tumoral activities through inhibition of EGFR tyrosine kinase activity. Accordingly they proposed to add gefinitib or erlotinib to VPA+doxo. Unfortunately, the proposed therapy was not tolerated in mouse model and they decided to use a multi target agent such as CUDC-101. The latter is an HDAC inhibitor that concurrently inhibit EGFR and HER2 activity. Conversely to doxo and VPA treatment, it is not clear how the authors choose the dose of 1uM as the right concentration of CUDC-101 for their experiment. Please provide a dose response curve and calculate IC50 value in H28, M14K and ZL34 cells treated with CUDC-101.

Please provide data of immunoblotting results observed in M14K and ZL34 relative to the phospho protein levels of EGFR, HER2, STAT1 and AKT. If possible, please also provide a representative blot of the pan EGFR, HER2,STAT1 and AKT protein levels for the above cited cells and for H28 cells reported in figure 4A.

Author Response

Comment 1: "... generate a doxo+VPA-resistante cell line. After that, authors should tested in this experimental condition their hypothesis, as well as checked whether a tgf ablation in this resistante cellular model could revert the acquired resistance...."

   This is indeed a very interesting suggestion that would extend the data presented in this paper. It is nevertheless unlikely that the involvement of a single factor (i.e. TGFa) in resistance to doxorubicin+VPA will be recapitulated by in vitro cell culture. As indicated by reviewer 2, other mechanisms such as multidrug resistance may indeed play a major role. The point that we would like to emphasize here is that there is a significant negative correlation between TGFa expression and sensitivity of MPM cell lines to VPA+doxorubicin (response to reviewer 2, Figure 1G and Supplementary Table S2). Importantly, the conclusion is also valid in patients since reduced expression of TGFa correlates with survival (Figure 5C) independently of the type of treatment and the response to treatment (stable disease, partial response and complete response; Figure 5E).

Comment 2: " Please provide a dose response curve and calculate IC50 value in H28, M14K and ZL34 cells treated with CUDC-101..."

   As requested we have added the dose response of the 3 cell lines to CUDC-101 (Figure S5). The dose (1mM) was chosen based on the metabolic activity (MTS) of the cells. We have added this information in the text (Result on line 159 and legend of Figure 4A and 4B). This dose was consistent with data reported in the literature obtained with other cell types [18,19].

   We added on line 159: "... The toxicity of CUDC-101 was evaluated by assessing the metabolic activity in H28, M14K and ZL34 cells (see dose response in MTS assay of Additional file: Figure S5). Based on these data and data from the literature [18,19], a subtoxic dose of 1 mM CUDC-101 was selected for further analysis..."

Comment 3: "... provide data of immunoblotting results relative to the phosphoprotein levels of EGFR, HER2, STAT1 and AKT. If possible, please also provide a representative blot of the pan protein levels..."

   We have now added a series of immunoblots in Supplementary Materials (ZL34, Figure S7). Due to the tight deadline allowed for answering to reviewers (from May 24 to June 2), we could not provide all immunoblots of pan protein levels.  

   We added on line 165: “… Intensity ratios as well as whole immunoblots including results observed in ZL34 cells can be found in Supplementary Materials (Additional files: Figures S6 and S7) …”

Reviewer 4 Report

Staumont et al. in the paper entitled “TGFα promotes chemoresistance of malignant pleural mesothelioma” examine the role of TGFα in promoting chemoresistance in MPM by studying gene and protein expression, apoptosis and the effects of gene silencing and transduction in two cell lines which show different responses to doxorubicin and VPA plus doxorubicin chemotherapeutic regimens.

Since long time it is known  that many MPMs show high expression levels of EGFR and a subset of cell lines express both EGFR and TGFα making both the receptor and downstream signaling pathways potential therapeutic targets. Stroumont et al. report quite convincing data showing that the H28 cell line, which is poorly responsive to doxorubicin and VPA plus doxorubicin, overexpresses TGFα. When TGFα expression is silenced, H28 cells become more responsive to the VPA plus doxorubicin treatment while a normally responsive cell line, M14K cells, transduced to overexpress the growth factor is desensitized to the apoptotic effect of the drugs. The synergistic effect of two TKIs and a three target molecule, CUDC-10, with the VPA plus doxorubicin or doxorubicin alone regimen has been also evaluated.

Concerns

Although data are well presented and convincing, I have a concern about some variability in % apoptotic  cells (annexine positive cells) reported in various figures, Fig. S2 to Fig. 2. Fig. 3, and Fig. 4. Were VPA and doxorubicin  used in Fig. 2 to 4 at the same concentrations as in Fig. S2. A certain variability can be intrisic to the type of assay and also due to different cell passage numbers. Were cells used within a passage number range?

Figure 1, panel 1: Was the ELISA perfomed just once? No SD is reported.

Author Response

Comment 1: "... some variability in % apoptotic cells (annexine positive cells) reported in various figures, Fig. S2 to Fig. 2. Fig. 3, and Fig. 4..."

   We do agree that the numbers of annexin-positive cells vary with the culture conditions and, in particular, with the dose of doxorubicin. In Figures 2, 3 and 4, the dose of doxorubicin was 0.1 mM as indicated in the M&M (line 370) while 0.5 mM were used in Figure S2 (mentioned in the figure legend). Another very important parameter is the selection of cells after transduction with lentivectors expressing targeted shRNAs (pLKO.1 shTGF) or expressing the TGFa (pCSEF-IB-TGFα or control pCSEF-IB). These cell lines were used in Figure 2 but not in Figures 3 and 4.

Comment 2:

   The ELISA was performed twice. Since the results were similar, only a representative analysis is shown.

Round 2

Reviewer 2 Report

Staumont et al. have significantly improved the manuscript with the changes made and answered to all this reviewer's comments in an appropriate manner. The manuscript is well written and will be of interest for the scientific community.

Reviewer 3 Report

Authors sufficiently answered to reviewer concerns